# A-Scan Ultrasonographic Evaluation of Patients with Idiopathic Intracranial Hypertension: Comparison of Optic Nerves

**DOI:** 10.3390/jcm11206153

**Published:** 2022-10-19

**Authors:** Nicola Rosa, Maddalena De Bernardo, Margherita Di Stasi, Ferdinando Cione, Ilaria Capaldo

**Affiliations:** 1Eye Unit, Department of Medicine, Surgery and Dentistry, Scuola Medica Salernitana, University of Salerno, Baronissi, 84081 Salerno, Italy; 2Radio Unit, Department of Medicine, Surgery and Dentistry, Scuola Medica Salernitana, University of Salerno, Via Salvador Allende, Baronissi, 84081 Salerno, Italy

**Keywords:** ocular ultrasound, idiopathic intracranial hypertension, interocular asymmetry, papilledema, optic nerve

## Abstract

Background: To evaluate the interocular optic nerve diameter (ONDs) asymmetry in patients with idiopathic intracranial hypertension (IIH) utilizing the A-scan ultrasound technique. Methods: Thirty-seven patients diagnosed with IIH were recruited from outpatients referred to the University Eye Unit between June 2014 and December 2021. Patients with optic disc pseudoedema or edema caused by other conditions were excluded. All patients with negative neuroimaging for intracranial space-occupying masses underwent standardized A-scan measurement of the OND in the primary gaze and lateral position (30 degrees test). Results: Mean, median, standard deviation, the minimum and maximum value of the two eyes at 0 degrees and the difference between the left and right thicker and thinner ONDs were measured. The two-tailed paired student *t*-test between the two eyes was performed using SPSS software. A statistically significant difference (*p*-value <0.001) between the two eyes, without a side prevalence, was found. Conclusions: Due to the differences between the ONDs of both eyes, we propose to use the mean of the ONDs between the left and right eyes at 0 degrees with the standardized A-scan diagnostic technique for a better follow-up of patients with IIH.

## 1. Introduction

Increased intracranial pressure (ICP) can be due to space-occupying lesions, such as tumors, and cerebral hemorrhages, but it can also be idiopathic [1].

Idiopathic intracranial hypertension (IIH) was originally described as serous meningitis by Quincke in 1893 [2]. Nonne described the syndrome more specifically coining the term “pseudotumor cerebri” in 1904 [3]. In 1955 Foley [4] attempted to simplify the nomenclature by defining it as “benign intracranial hypertension”. However, in view of the potential vision loss associated with papilledema, Corbett and Thompson replaced the adjective “benign” with “idiopathic” to better highlight the absence of an underlying organic pathological condition [5].

IIH mainly affects overweight women, especially those of childbearing age, with an estimated prevalence of nearly 1 case in every 100,000 women, but this increases to 13 cases every 100,000 in those of 20 to 44 years, mainly if they are above 10% of their ideal body weight [6]. Men are less frequently affected, with a female:male ratio of 4–8:1 [7].

In case of IIH, to detect changes in ICP is very important. Papilledema has been proven not to be very sensitive for such a purpose because, with the increase and decrease in ICP, it can take days to appear and disappear [8].

A lumbar puncture is considered the most reliable way to monitor ICP, but this method is quite invasive, and cannot be used as a routine examination. For this reason, the measurement of the optic nerve (ON) diameter has been suggested as a sensitive and reliable method for following these patients. ICP rise will lead to a subarachnoidal fluid increase in the ON sheaths that can be quite easily detected by measuring its diameter with ultrasound [9,10,11,12,13].

Theoretically, in the case of an ICP rise, a bilateral ON increase should be present, but it has not been clearly shown.

To be able to detect an ICP increase or decrease which could influence the therapeutic decision during follow-up is crucial.

To the best of our knowledge, only one paper, utilizing B-scan echography, looked for such symmetry, and an asymmetric appearance of the ON diameter was found [14].

Unfortunately, B-scan echography did not prove to be very reliable in measuring the ON diameter due to several reasons including the so-called “blooming effect”, among others [12,13,15].

In light of these observations, we decided to check if this asymmetry is also present with the so-called “standardized” A-scan technique, or if the reported asymmetry is related to the poor reliability of this technique [13,16].

## 2. Materials and Methods

Patients referred to the University Eye Unit between June 2014 and December 2021 with a diagnosis of IIH, based on the modified Dandy’s criteria, were evaluated [1].

IRB approval was obtained from Cometico Campania Sud (CECS) (Prot. Number 16544).

Informed consent was obtained from each patient. Patients with optic disc pseudoedema (drusen, tilted disc, myelinated fibers) or edema caused by other conditions, such as ON tumors, inflammation, brain tumors or cerebral hemorrhages were excluded.

The included patients underwent ON diameter measurement with standardized A-scan echography [13].

All examinations were performed by an expert examiner in ophthalmic ultrasound (NR) using the Cinescan S and ABSolu systems (Quantel Medical, Bozeman, MT) utilizing an unfocused 8 MHz standardized A-scan probe. The measurements were obtained by placing the probe on the temporal bulbar conjunctiva, previously anesthetized with oxybuprocaine hydrochloride eye drops with the patients lying on their back and with their eyes in a primary gaze position.

Through fine adjustment of the probe’s angle, the examiner obtained two extremely high and steeply rising spikes coming from the arachnoidal inner surfaces (Figure 1 and Figure 2).

If the ON exceeded the clinically established normal limit (4.5 mm), the patient was instructed to abduct the eye (so-called “30 degrees test”) and the measurement was repeated in both eyes. The percentage of ON reduction measured in the lateral gaze position compared to the primary position was calculated. A reduction greater than 10%, suggesting increased fluid around the ON, was considered diagnostic for IIH [13].

Descriptive statistics and statistical analysis were performed with SPSS software (Version 26.0, SPSS Inc., Chicago, IL, USA). The normality of data was checked with the exact Kolmogorov–Smirnov: all data were normally distributed (all *p* > 0.05). The following parameters regarding OND were calculated:-Mean,-Median,-Standard deviation,-Minimum and maximum.

Differences between intraindividual thicker and thinner OND eyes were also evaluated, i.e., the mean value of differences was the primary outcome.

Due to the normal distribution of all analyzed data, a paired student *t*-test was performed to compare subgroups. For each subject, the asymmetry was evaluated at 0 degrees between the two eyes, without taking into account the 30 degrees measurements.

## 3. Results

Thirty-seven caucasian patients (12 males and 25 females) with diagnoses of IIH and with a mean age of 27.19 ± 15.22 years (range 6.81 to 65.86 years) were recruited.

The patients’ data as mean, median, standard deviation, minimum and maximum differences between right and left eye ONs are shown in Table 1.

Forty-two healthy caucasian patients (13 males and 29 females) with a mean age of 30.88 ± 3.55 years (range 5.58 to 68.92 years) were utilized as a control group.

The control group’s data as mean, median, standard deviation, minimum and maximum values of the differences between right and left eye ON are shown in Table 2.

The same data of the two groups, as the difference between interocular thicker and thinner ONs, is shown in Table 3.

The results demonstrate a statistically significant difference between the two eyes (*p* < 0.001) without a predominance between the left and right side (Figure 3) in both groups, meaning that an asymmetry is present in both healthy patients and the presence of IIH, but the mean of the interocular difference in case of IIH (0.58) is larger compared to the control group (0.31).

## 4. Discussion

The results of the present study highlighted a statistically significant difference between the two eyes (*p* < 0.001) without a side prevalence, in both groups.

Several hypotheses could explain the presence of asymmetric ON measurements both in the case of IIH and control, including the size of the optic canal, optic atrophy, fluid composition and ocular pressure, and changes in lamina cribrosa or ON structures among others [17].

The size of the optic canal could play a crucial role in the CSF flow dynamics [18]. A smaller canal could transmit the CSF pressure along the ON less easily, thus determining less of an increase in the ON diameter [19]. However, data on optic canal diameter in patients with IIH are needed to better under-stand whether this size asymmetry is congenital or may result from long-lasting CSF-related bone erosion, as described elsewhere in the skull base of patients with chronic IIH [20].

Several papers describe the use of ultrasound to determine or follow up the intracranial pressure [21,22,23,24], but only a few of them focused on a difference between left and right ONs with conflicting results [14,25,26,27].

Naldi A. et al. [14] examined the OND in 40 healthy subjects and 40 patients with IIH and found an interocular asymmetry in both groups.

On the contrary, other studies were not able to find the difference in ON sheath diameter between the left and right eyes of patients with IIH [25,26,27].

Our findings could explain why these studies reached different conclusions; in fact, we too found no significant differences between right and left eyes when we compared their mean values, but we found differences when we compared interocular differences obtained in each patient. This means that the difference is present, but there is not a prevalence of the left over the right and vice versa.

However, the main limitation of the previous studies is that they have been performed with B-scan which has been widely criticized due to the presence of several artifacts [28].

Among the main B-Scan-related problems, there is the lack of a standard system sensitivity setting. This will make it very difficult to compare different images because images with the lower setting will show larger ONDs (so-called “blooming effect”), making it impossible to realize a precise normality range [29]. To overcome this problem, for each instrument, the same gain setting should be utilized with all patients. This could allow a specific normal range for every single device to be defined, but this is very difficult to achieve because it would require a very large number of patients, making it not that suitable [30].

Unfortunately, the blooming effect is not the only problem related to the use of B-scan devices. As shown by Coppetti and Cattarossi [15], what is generally measured with B-scan is just an artifact, but even taking into account their advice to image the central retinal artery with color Doppler, there are other problems such as the scattering related to the echo beam reaching the sheets in a non-perpendicular way, which cannot be overcome.

For this reason, to prevent these artifacts, we utilized the so-called standardized A-scan technique which is more sensitive than the B-scan technique when dealing with less than 0.5 mm, as it so occurs when evaluating the ON sheath. Moreover, it allows the so-called “30 degrees test” to be performed which can differentiate increased OND due to the increase in the subarachnoidal fluid from solid lesions, such as tumors or inflammations [13].

A limitation of this study could be represented by the small sample size; therefore, further studies on different and larger samples to determine the effectiveness of the OND mean value parameter should be performed to evaluate patients with IIH.

In conclusion, in our study, a significant difference between the left and right eye, without a predominance of side, was highlighted.

This finding confirms that the presence of an asymmetry in the ONDs is real and not due to the unreliability of the B-scan technique, making the measurement of both ONs and using the mean measurements in detecting changes in ICP during the follow-up visits mandatory.

Therefore, due to the other B-scan technique limitations, the mean of the ONDs between the left and right eyes at 0 degrees with the standardized A-scan diagnostic technique should be used for a better follow-up of patients diagnosed with IIH.

## Figures and Tables

**Figure 1 jcm-11-06153-f001:**
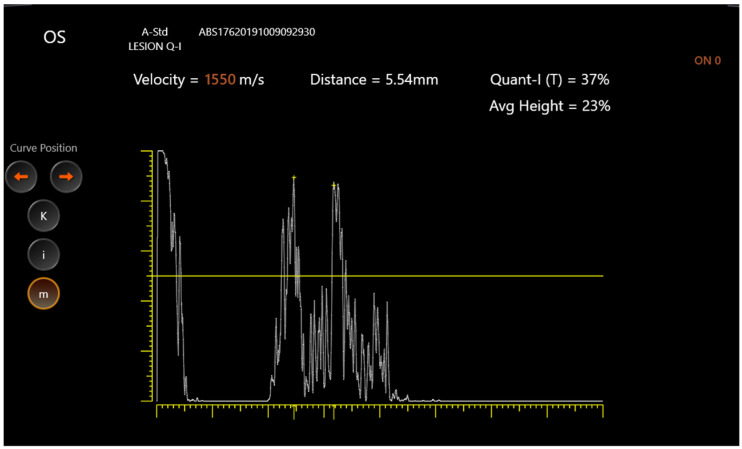
Optic nerve diameter ultrasound. Standardized A-scan ultrasound showing left optic nerve diameter at 0 degrees.

**Figure 2 jcm-11-06153-f002:**
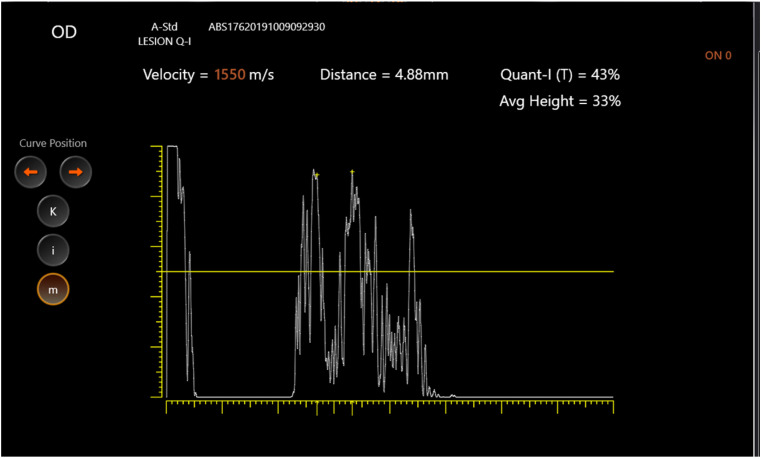
Optic nerve diameter ultrasound. Standardized A-scan ultrasound showing right optic nerve diameter at 0 degrees.

**Figure 3 jcm-11-06153-f003:**
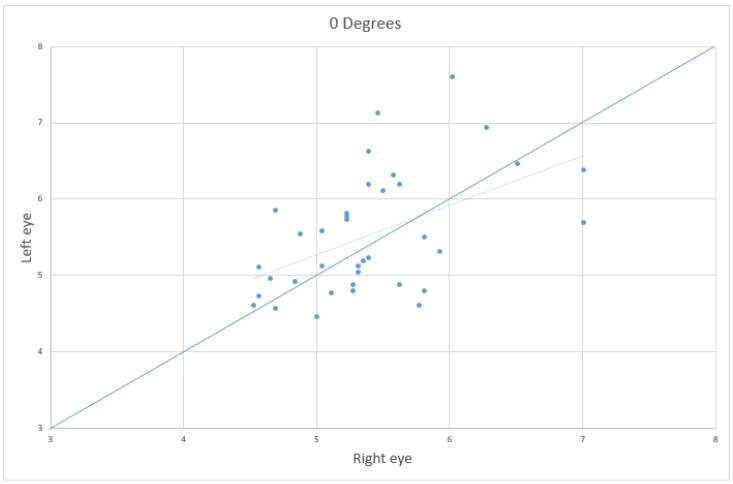
Scatter plot showing the correlation between the right (“x” axis) and left (“y” axis) optic nerve diameter in millimeters, with the eye in a straight gaze position.

**Table 1 jcm-11-06153-t001:** Left and right ONDs in a straight gaze in patients with IIH (0 degrees).

OND 0 Degrees		
	Right Eye (mm)	Left Eye (mm)
Mean	5.40	5.53
Median	5.31	5.31
Standard deviation	0.61	0.79
Minimum	4.53	4.46
Maximum	7.01	7.6
Test-T	0.29	

OND: Optic nerve diameter. IIH: Idiopathic intracranial hypertension.

**Table 2 jcm-11-06153-t002:** Left and right ONDs in a straight gaze in patients of the control group (0 degrees).

OND 0 Degrees		
	Right Eye (mm)	Left Eye (mm)
Mean	3.5	3.5
Median	3.5	3.5
Standard deviation	0.5	0.4
Minimum	2.2	2.6
Maximum	4.5	4.3
Test-T	0.69	

OND: Optic nerve diameter.

**Table 3 jcm-11-06153-t003:** Differences between thicker and thinner OND in patients with IIH and the control group.

OND Right–Left Eye (mm)		
	IIH	Control Group
Mean	0.58	0.31
Median	0.54	0.31
Standard deviation	0.42	0.25
Minimum	0.04	0.00
Maximum	1.67	1.11
Test-T	<0.001	<0.001

OND: Optic nerve diameter. IIH: Idiopathic intracranial hypertension.

## Data Availability

The data presented in this study are available on request from the corresponding author.

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
