# Peer review of "A-Scan Ultrasonographic Evaluation of Patients with Idiopathic Intracranial Hypertension: Comparison of Optic Nerves"

_jcm, 2022, doi:10.3390/jcm11206153_

Round 1

Reviewer 1 Report

Dear Authors,

I wish to submit my review for the paper titled: 

"A-SCAN ULTRASONOGRAPHIC EVALUATION OF PATIENTS WITH IDIOPATHIC INTRACRANIAL HYPERTENSION: COMPARISON OF OPTIC NERVES"

The Subject is novel and the authors should be commended for their work. Despite the novelty, Some points require to be proofread:

Could you please reformulate the Abstract conclusions?

Could you please add the reference for the article mentioned in the introduction? Page 2: lines 63-64

Could you please write a specific subparagraph for statistical analysis? Please expand it, specifying how the quantitative and qualitative variables were presented across the studies. (e.g., average, median, and standard deviation)

Have you considered a regression model to analyze the different covariates' role in the ONDs asymmetry?

Discussion:

Some sentences require English language proofreading. The are no limitations in the study. In addition,

Could you please start the discussion with a sentence explaining the overall findings?

Finally, Could you please amend and expand it according to your results? 

Reviewer 2 Report

This is an interesting study reporting the measurements of the ON in case of IIH. I have no specific questions or comments pertaining to this manuscript. I might suggest the Authors to have the text be revised by an English native speaker to brush up some syntax errors in the manuscript.

Reviewer 3 Report

The introduction is insufficient, and the novelty and significance of this paper is not clear.

The workload of this paper is far from meeting the requirements of journal papers.

Please provide the demographic information.

Control group is missing.

Please improve the quality of figures.

Reviewer 4 Report

Thank you for allowing me to review this interesting paper.

My comments are:

1.    There is no control group.

2.    The article does not mention OCT (Optical Coherence Tomography).

3.    What is the advantage of measuring optic nerve diameter by A-Scan Ultrasonography over optic nerve head measurements by RNFL OCT? RNFL OCT can also be compared between both eyes and is easier to perform by less qualified staff than A-Scan Ultrasonography.

4.    The age range of the patients is very wide and atypical for IIH (6.81-65.86 years). Please explain.

5.    The conclusion is not clear to me: " the mean of the ONDs between the right and left eyes at 0 degrees with the standardized A-scan diagnostic technique could be used for a better follow-up of patients diagnosed with IIH". How exactly? Will each ON diameter be measured in each visit and compared to itself?  Will the difference between both OND be measured at each visit and compared from visit to visit? How could this technique be used for a better follow-up of patients diagnosed with IIH?

Round 2

Reviewer 3 Report

I don't believe this paper can be a qualified academic paper or even a lab report.

Author Response

thank you for your comment

Reviewer 4 Report

All my comments have been fully addressed. 

Good luck!

Author Response

Thank you for your suggestions to improve the manuscript
